# Micromechanical Effect of Martensite Attributes on Forming Limits of Dual-Phase Steels Investigated by Crystal Plasticity-Based Numerical Simulations

Tarek Hussein [1], Muhammad Umar [1,2,*], Faisal Qayyum [1], Sergey Guk [1] and Ulrich Prahl [1]

1   Institute of Metal Forming, Technische Universität Bergakademie Freiberg, 09599 Freiberg, Germany; tarekelsesi22@gmail.com (T.H.); faisal.qayyum@imf.tu-freiberg.de (F.Q.); sergey.guk@imf.tu-freiberg.de (S.G.); ulrich.prahl@imf.tu-freiberg.de (U.P.)
2   Department of Mechanical Engineering, Khwaja Fareed University of Engineering and Information Technology, Rahim Yar Khan 64200, Punjab, Pakistan
*   Correspondence: umar.matrix@gmail.com

**Abstract:** This study analyses the effect of martensite grain size and its volume fraction in dual-phase (DP) steel on (1) the formability limit, (2) average global behavior under different loading conditions, and (3) damage initiation. The virtual RVEs (Representative Volume Elements) were constructed using DREAM.3D software with a variation of microstructural attributes. The numerical simulations were carried out using DAMASK, which evaluates the polycrystalline material point behavior and solves versatile constitutive equations using a spectral solver. The simulations were post-processed to obtain global and local stress, strain, and damage evolution in constructed RVEs. The global results were processed to obtain FLDs according to Keeler-Brazier (K-B) and Marciniak and Kuczynski (M-K) criteria. In this work, the capability of microstructure-based numerical simulations to analyze the FLDs has been established successfully. From Forming Limit Diagrams (FLDs), it was observed that formability changes by changing the strain hardening coefficients (n-values), the martensite fraction, and martensite grain sizes of DP steels. The improved formability was observed with lower martensite fraction, i.e., 17%, decreased martensite grain size, i.e., 2.6 μm, and higher strain hardening coefficient. The M-K approach shows the better capability to predict the formability by various loading conditions and clarifies the necking marginal zone of FLD. The damage propagation is also strongly affected by the loading conditions. The current study would be a good guide for designers during the manufacturing and selecting of appropriate DP steels based on the service loading conditions.

**Keywords:** dual-phase steel; forming limit diagrams; crystal plasticity; DAMASK; M-K approach; Keeler-Brazier approach

## 1. Introduction

Dual-Phase (DP) steel, due to its higher energy absorption capacity and reduced weight, is used in the automotive industry to achieve simultaneous high strength and elongation goals [1–3]. Hard phase martensite laths embedded in the softer ferrite matrix are responsible for reinforcing the solid aggregate, while ductility is incorporated by the matrix [4]. Dual-phase steel is a suitable example of a multi-phase material because of the significant difference in the mechanical properties of its phases, and it has widespread usage in the automotive industry. This peculiar combination of hard and soft phases imparts desirable properties in the material, i.e., low 0.2% proof stress and a high work-hardening coefficient (n-value). Generally, during cold forming processes of drawing and stretching, higher n-values exhibit uniform global formability by avoiding local thinning [5]. However, practical aspects often reveal unexpected necking and local failure in complex forming during bending or flanging processes. Due to the heterogeneous microstructure of soft

ferrite and hard martensite, local straining causes unpredicted local abnormalities. The heterogeneity affects the micro-scale attributes of materials. Consequently, it influences the component scale's material properties, particularly the material damage behavior [6]. Therefore, it is of the utmost importance to investigate the relationship between the phases' heterogeneity and their microstructural attributes, especially martensite and ferrite fractions and their grain sizes [7].

Material formability is a fundamental mechanical property for vehicle bodies and structural members in the automotive sector. Therefore, accurate predictions of its indicators and reliable utilization are needed. For this purpose, FLDs are commonly used to estimate sheet metal formability. Its purpose is to predict safe, necking, and failure zones by adopting major and minor forming limit strain inside a diagram [8]. Experimentally, the Nakajima test detects these forming limit strains, which is used by performing a punch–die method on specimens with varying dimensions under different loading conditions. The test shows a higher accuracy for most materials, but it is expensive, slow, and relies on a complex specimen-shaping process. Therefore, FLDs are plotted using some well-established numerical models [9]. Tasan et al. [10] carried out nanoindentation experiments and concluded that the numerical damage could not be applied with better accuracy to the martensite phase because the indents are the same size or bigger than martensite grains. While there is a considerable difference in the indent and average grain size for ferrite, the identified parameters in their study could predict the damage in the ferrite phase [11].

Analytical models commonly applied to predict and plot FLDs by adopting major and minor limit strains are Swift, Chow–Hosford, Keeler–Brazier, and Marciniak–Kuczynski [11]. These approaches have been analyzed and compared in depth in research work by Basak and G Béres [12,13]. These approaches depend on some variables to develop the mathematical model and corresponding limit strains, i.e., Swift models require strain ratios and strain hardening coefficient values. Chow equations depend on the anisotropy coefficients, and for the Keeler–Brazier model, the thickness value and strain hardening coefficient are primarily important. The simulations in this study were performed using two approaches, i.e., the Keeler–Brazier, and M-K approaches, because of their efficiency, popularity, and simplicity in plotting FLDs of high-strength steels recommended by Duancheng Ma, Basak, and Béres. Kuang-Hua Chang et al. [14] used damage percentage detection to show the marginal zone of an FLD, where the area under this zone is safe, while the area above it is a damage zone.

Virtual modeling and simulation tools for the sheet metal processes play an essential role in predicting mechanical behavior and have become an essential and inevitable part of each industry [15,16]. These tools introduce full behavior prediction models of the metals, starting from their production to the heat treatment and testing processes. It helps engineers and designers to reduce the resource-consuming experiments and push the overall economic and technical aspects forward. A significant amount of work is being carried out toward developing, validating, and implementing numerical models to improve the accuracy of the simulation results. The numerical simulations method is an intelligent tool used to solve complex equations with different variables, called constitutive equations. These equations and variables represent the physical situations of the materials, e.g., deformation mechanisms, and help analyze the phenomena of mechanical deformation, damage, and failure. It can also predict crucial mechanical thresholds for high-end application materials during melting, casting, forming, and machining [17].

Hutchinson [18] built a model for FCC crystal, which was later expanded for BCC and HCP crystal structure and applied to DAMASK (Düsseldorf Advanced Materials Simulation Kit). Michel et al. [19] stated that the heterogeneity of ferrite and martensite phases in DP steels and their elastic stiffness coefficients play an essential role in the convergence behavior and stability schemes. After that, Diehl et al. [20] used some FFT assumptions and developed the stiffness coefficients applied later in the DAMASK framework. [7] The phenomenological crystal plasticity model is employed as plasticity law, which assumes that plastic deformation occurs on a slip system when the resolved shear stress exceeds

a critical value. The resolved critical shear stress depends on the amount of the applied stress and can be calculated from a relation known as Schmid's law in Equation (1) [19,21].

$$\dot{\gamma}^{\alpha} = \dot{\gamma}_0 \left| \frac{\tau^{\alpha}}{S^{\alpha}} \right|^{n} \text{sgn}(\tau^{\alpha}) \tag{1}$$

Analytical techniques can merely solve partial differential equations used in these constitutive models more straightforwardly, while numerical methods are essential for complex forms. Many efforts have been made to reach a framework connecting these boundary problems with physical phenomena [18]. Many numerical methods, i.e., Finite Element Method (FEM), Finite Volume Method (FVM), spectral method, and the Fast Fourier Transform (FFT)-based (Crystal Plasticity Finite Element Method) CPFEM method are usually used [22,23]. The difference between FE and SP methods lies in their homogenization technique and consequent time saving, as Shanthraj et al. [24] reported. The analytical FLD strain calculation models must be solved numerically with a high-efficiency solver to construct FLDs [25].

A representative volume element (RVE) was constructed for this purpose as a virtual sample to express the properties and microstructure of the material in the sample. In this study, spectral solver methods were applied using Fast Fourier Transform (FFT) to solve the boundary value problem for mechanical equilibrium and damage phase field. An FFT-based spectral solver shows higher efficiency and faster computational time over the domain.

The elastic and plastic material properties are governed by the crystal plasticity-based constitutive model, which is extensively used to study the deformation in crystalline materials. Besides considering microstructural parameters, it also considers some fitting parameters to compensate for the influence of some complex phenomena happening during the processing route of a specific grade of the material [25,26]. The simulations presented in this work were performed using DAMASK, which is available as free and open-source software [25]. It aims to simulate the material using crystal plasticity principles within a finite strain framework for continuum mechanical considerations by modeling the material point (Fourier point) inside the constructed mesh. The plasticity laws implemented in DAMASK are represented by isotropic plasticity, phenomenological crystal plasticity, or dislocation density-based crystal plasticity. The phenomenological crystal plasticity models were used in these simulations of polycrystal models. Furthermore, as deformation of polycrystalline aggregate strongly depends on the respective orientations of grains inside the lattice, consideration of Euler angles and rotation matrices was taken during the implementation of the model [20,25].

In this work, the establishment of a crystal plasticity-based approach to evaluate the forming limits of multi-phase steels by numerical simulation has been carried out successfully. A plastic instability approach and a high-efficiency numerical solver were employed to investigate multi-phase materials' formability by adopting major and minor strains in FLDs. Specifically, the microstructural attributes of martensite, i.e., grain size and phase fraction under different loading conditions, were studied. Furthermore, the damage initiation and propagation were determined by a CPFFT-based spectral solver using a phenomenological model for plastic deformation and Hooke's law for elasticity. The conclusions of this study can help the sheet metal industries as a guide for the designers to process material in an improved way.

## 2. Methodology

The numerical simulation modeling approach is a systematic process that uses well-defined pre-processing steps, running a set of simulations, post-processing, and visualization of the results, as shown in Figure 1. Firstly, RVEs were constructed by varying martensite volume fraction and martensite grain size by using DREAM.3D [27]. Next, the DAMASK adopts individual grains details to run numerical simulations using given load and geometry configuration files. Constitutive equations and microstructural attributes

for ferrite and martensite phases were adopted from Tasan's work [10]. Ductile damage and degradation parameters already implemented by Shanthraj et al. [24] were applied to the ferrite phase within a pre-developed model by Roters and Tasan et al. [20,25]. Finally, the values of local and global results were extracted after simulations and visualized by ParaView and other trend-plotting tools.

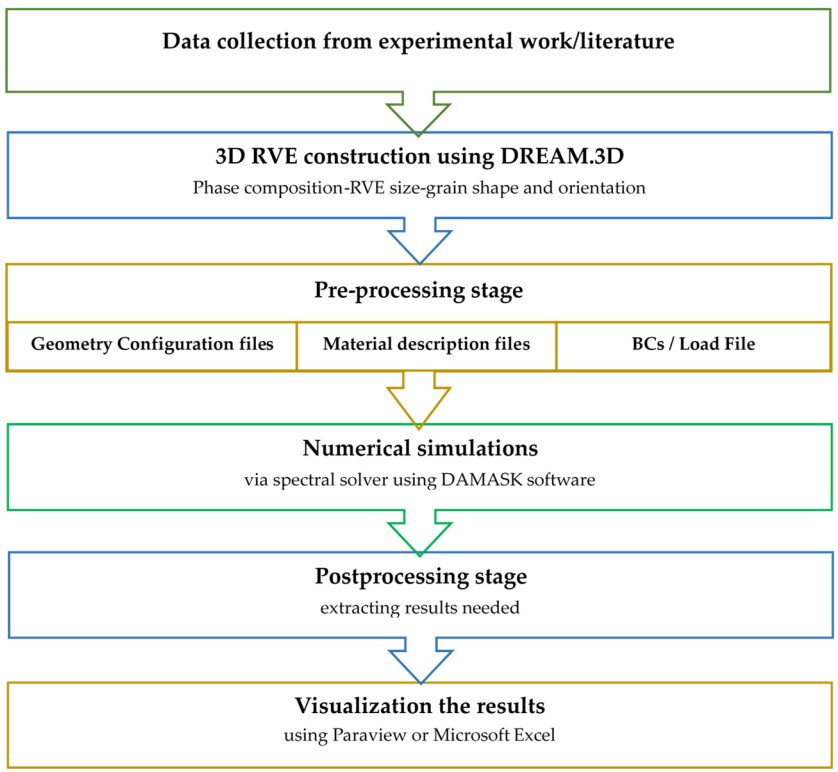

**Figure 1.** A simple flow chart showing the methodology adopted and data flow in this study.

### 2.1. Data Collection

The microstructural attribute values, e.g., martensite fractions and grain sizes, were taken from Tasan et al. [10], as shown in Table 1. The study was carried out for DP600 with the variation of martensite attributes (fractions 1.2%, martensite grain size from 1.0 to 4.3 μm) and a considerable ferrite grain size difference within 2.2 to 14.5 μm.

**Table 1.** The chemical composition and microstructural characteristics of DP 600 used in the current study. Reprinted with permission from Ref. [10] Copyright 2014 Elsevier.

| Steel | Martensite (%) | Ferrite Grain Size (μm) | Martensite Grain Sizes (μm) |
|---|---|---|---|
| DP600 | 17.2 | 8.4 ± 6.1 | 2.7 ± 1.6 |
| | 18.4 | 4.9 ± 1.9 | 1.7 ± 1.1 |

### 2.2. RVE Construction

From DP steel values in Table 1, four RVE models (A–D) were constructed using DREAM.3D with variations in RVE dimensions, phase fractions, martensite grain size, and spatial distribution (refer to Figure 2). However, in all the five RVE models, the same hardness properties of martensite and ferrite grain size were considered, as shown in Table 2.

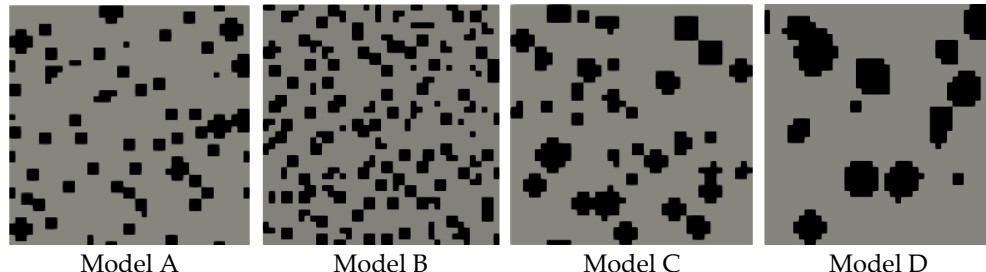

| Model A | Model B | Model C | Model D |

**Figure 2.** Top surfaces of 3D RVE models of the dual-phase steels, where the black particles represent the martensite phase inside the grey ferrite matrix.

**Table 2.** The projected RVEs used in the study with ferrite grain size $8.4 \pm 6.1$ µm for all the RVEs.

| Parameter | Model A | Model B | Model C | Model D |
|---|---|---|---|---|
| Synthetic volume size (voxels) | $40 \times 40 \times 10$ | $40 \times 40 \times 10$ | $40 \times 40 \times 10$ | $40 \times 40 \times 10$ |
| Martensite grain size (µm) | $2.7 \pm 1.6$ | $2.7 \pm 1.6$ | $4 \pm 2$ | $6 \pm 2$ |
| Martensite volume fraction | 17% | 18% | 17% | 17% |
| Ferrite volume fraction | 83% | 82% | 83% | 83% |

Suitable statistical distributions, cubic crystal structure, and ellipsoid grain shapes closely mimic the actual microstructure of DP Steel. In addition, different dimensions of synthetic volume were adopted to check the effect of RVE size on the global stress–strain curves and FLDs. Qayyum et al. [28] have shown a more detailed framework for the RVE generation using the DREAM.3D pipeline in their work. If interested, the readers are encouraged to refer to their work for further details.

*2.3. Pre-Processing Stage*

2.3.1. Material Properties

The ferrite and martensite phases in DP steel have some common elastic–viscoplastic properties, which help build a more manageable material file framework, despite variations in their mechanical behavior and properties. The material parameters and damage values of both phases were adopted from Qayyum et al. [7], wherein already developed and validated models from the framework of DAMASK [25] were adopted. The elastic coefficients, initial and saturated shear resistances of slip systems and fitting parameters from the already published literature [10,29] were used as presented in Table 3. Regarding damage, the already developed models from Roters et al. [25] were incorporated and adopted in the material configuration files of respective RVEs. The critical plastic strain value $\epsilon_{\text{crit}}$ for the ferrite phase was taken as 0.5 [10].

2.3.2. Boundary/Loading Conditions

Different loading conditions were used in the simulation process with four strain states on each RVE model along y-directions while controlling x-directions and freeing the z-directions (refer to Figure 3).

The periodic boundary conditions were stated as:

In the uniaxial tension state:

$$\dot{\mathbf{F}} = \begin{bmatrix} * & 0 & 0 \\ 0 & 1 & 0 \\ 0 & 0 & * \end{bmatrix} \times 10^{-3} \text{ s}^{-1} \qquad \& \qquad \mathbf{P} = \begin{bmatrix} 0 & * & * \\ * & * & * \\ * & * & 0 \end{bmatrix} \text{Pa} \qquad (2)$$

**Table 3.** Physical and fitting parameter values were adopted from the published literature for ferrite [29], reprinted with permission from Ref. [10] Copyright 2014 Elsevier, and martensite [7].

| Parameter Definition | Symbol | Ferrite Attributes | Martensite Attributes |
|---|---|---|---|
| Lattice crystal structure | lattice structure | bcc | bcc |
| First elastic stiffness constant with normal strain | C11 | 233.3 GPa | 417.4 GPa |
| Second elastic stiffness constant with normal strain | C12 | 135.5 GPa | 242.4 GPa |
| First elastic stiffness constant with shear strain | C44 | 118.0 GPa | 211.1 GPa |
| Shear strain rate | $\gamma^\alpha$ | $10^{-3}/s$ | $10^{-3}/s$ |
| Initial shear resistance on [111] | $S_o$ [111] | 95 MPa | 405.8 MPa |
| Saturation shear resistance on [111] | $S_\infty$ [111] | 222 MPa | 872.9 MPa |
| Initial shear resistance on [112] | $S_o$ [112] | 97 MPa | 456.7 MPa |
| Saturation shear resistance on [112] | $S_\infty$ [112] | 412 MPa | 971.2 MPa |
| Slip hardening parameter | $h_0$ | 1000 MPa | 563.0 MPa |
| Interaction hardening parameter | $h_{\alpha,\beta}$ | 1 | 1 |
| Stress exponent | n | 20 | 20 |
| Curve fitting parameter | w | 2 | 2 |
| Damage parameters | | | |
| Interface energy | $g_0$ | $1.0 \text{ J m}^{-2}$ | - |
| Characteristic length | $l_0$ | 1.5 μm | - |
| Damage mobility | M | $0.01 \text{ s}^{-1}$ | - |
| Damage diffusion | D | 1.0 | - |
| Critical plastic strain | $\epsilon_{crit}$ | 0.5 | - |
| Damage rate sensitivity | P | 10 | - |

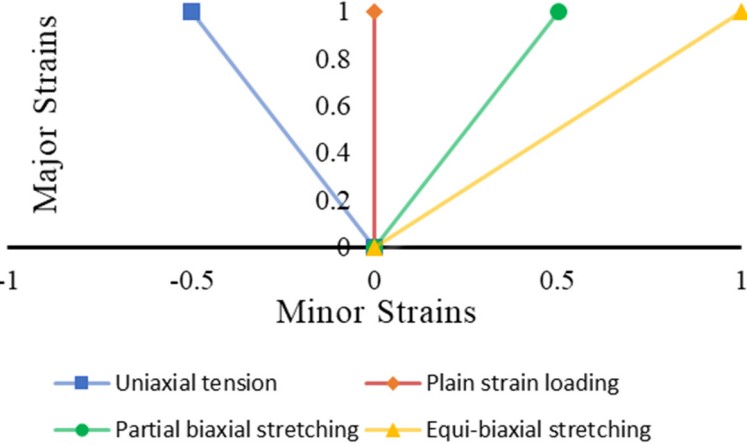

**Figure 3.** Projected loading states to conclude the forming limit diagram.

In the plane strain state:

$$\dot{\mathbf{F}} = \begin{bmatrix} 0 & 0 & 0 \\ 0 & 1 & 0 \\ 0 & 0 & * \end{bmatrix} \times 10^{-3} \text{ s}^{-1} \qquad \& \qquad \mathbf{P} = \begin{bmatrix} * & * & * \\ * & * & * \\ * & * & 0 \end{bmatrix} \text{Pa} \qquad (3)$$

In the partial biaxial stretching state:

$$\dot{\mathbf{F}} = \begin{bmatrix} 0.5 & 0 & 0 \\ 0 & 1 & 0 \\ 0 & 0 & * \end{bmatrix} \times 10^{-3} \text{ s}^{-1} \qquad \& \qquad \mathbf{P} = \begin{bmatrix} * & * & * \\ * & * & * \\ * & * & 0 \end{bmatrix} \text{Pa} \qquad (4)$$

In the equi-biaxial stretching state:

$$\dot{\mathbf{F}} = \begin{bmatrix} 1 & 0 & 0 \\ 0 & 1 & 0 \\ 0 & 0 & * \end{bmatrix} \times 10^{-3}\ \mathrm{s}^{-1} \qquad \& \qquad \mathbf{P} = \begin{bmatrix} * & * & * \\ * & * & * \\ * & * & 0 \end{bmatrix}\ \mathrm{Pa} \qquad (5)$$

where $\dot{\mathbf{F}}$ and $\mathbf{P}$ are the rate of the deformation gradient and first Piola–Kirchhoff stress tensors, respectively. The coefficients denoted by '$*$' express the stated complimentary conditions. Using these conditions, the four strain states were applied in the y-direction, with a $1 \times 10^{-3}\ \mathrm{s}^{-1}$ iso-static strain rate, as shown in Figure 3. For readers not familiar with the modeling strategy, a brief model background is provided in the work of Qayyum et al. [7] and Duancheng Ma et al. [30].

### 2.4. Evaluation of FLDs by M-K and K-B Approaches

### 2.4.1. M-K Approach

In this approach, the engineering stress and strain values were extracted by applying customized subroutines. Then engineering stress–strain curves of the four loading conditions for each RVE were plotted. According to Duancheng Ma [30], Drucker's stability criterion claims that the forming limit occurs at the maximum stress (localized necking point) on the engineering stress–strain curve.

### 2.4.2. Keeler-Brazier Approach

In this approach, the true stress and strain values were extracted from applying already customized subroutines, and then the true stress–strain curve was plotted. As per the Keeler–Brazier model detailed in appendix A, the major strain values ($\varepsilon_1$) depend on t (RVE thickness), n (strain hardening coefficient of material), and $\varepsilon_2$ (minor strain values detected from the tensor matrix at an increment, corresponding to the maximum stress on the stress and strain curve). Therefore, the major and minor strain values were available for the four loading conditions. These and other extended details about post-processing and an activity flow chart of M-K and K-B approaches are given in Appendix A.

## 3. Results

Simulations of multi-phase DP steel were processed for 3D RVEs with damage consideration, wherein four models were virtually constructed to detect the variation in FLDs' behavior by varying microstructural attributes. By simulating different synthetic volumes of RVEs with the same ferrite grain sizes, the varying behavior of engineering stress–strain curves and FLDs was observed.

### 3.1. Effect of Martensite Fractions on Stress–Strain Curve and FLDs

The numerical simulations were carried out for models A and B as per Figure 2, and the difference in martensite volume fractions, as shown in Table 2. The behavior of stress–strain curves of uniaxial, plane strain, biaxial, and equi-biaxial loading is shown in Figure 4. By increasing martensite volume fractions, the yield and ultimate strengths increased while the values of corresponding engineering strains decreased. The stress values of uniaxial loading were almost like those of plane strain loading conditions but much lower than biaxial and equi-biaxial loading conditions. By calculating areas under engineering flow curves for the four loading conditions, the plastic work per unit volume increased by decreasing the martensite fraction in DP steels. The plastic work performed by the biaxial loading case ranged between 37 and 39 MJ/m$^3$, while uniaxial and plane strain loading cases ranged from 28 to 36 MJ/m$^3$.

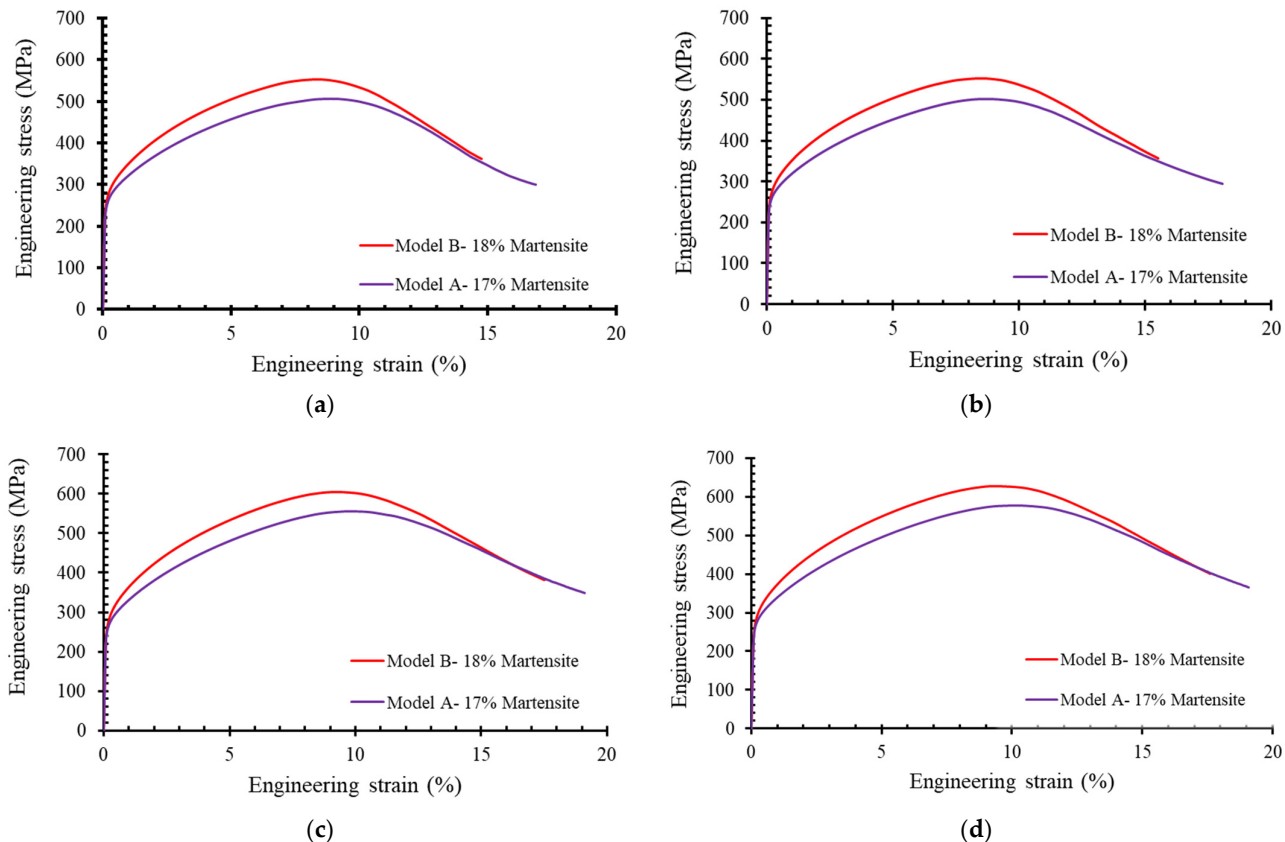

**Figure 4.** Comparison of stress–strain curves of different martensite volume fractions in (**a**) uniaxial tension, (**b**) plane strain tension, (**c**) partial biaxial tension, and (**d**) equi-biaxial loading.

Concerning FLDs, the results are more significant, as shown in Figure 5. It was observed that the increase of martensite volume fractions decreased the values of major and minor strains of FLDs slightly in the M-K approach. However, in the case of the K-B approach, the formability difference was not observed.

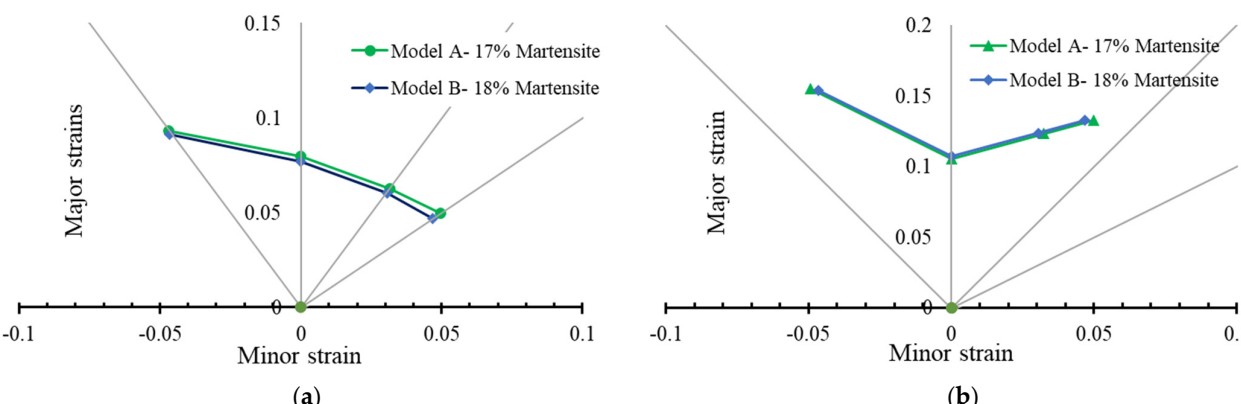

**Figure 5.** The FLDs in martensite fraction differences in the case of (**a**) M-K approach and (**b**) K-B approach.

By plotting the logarithmic true stresses and strains of models (A and B), both variables were directly proportional to each other; when the values of n-values were detected from the slope of the graph. It was found that by decreasing the martensite volume fractions, the values of n-values increased, as shown in Figure 6 and Table 4. The trend is similar to what is already reported in the literature [25].

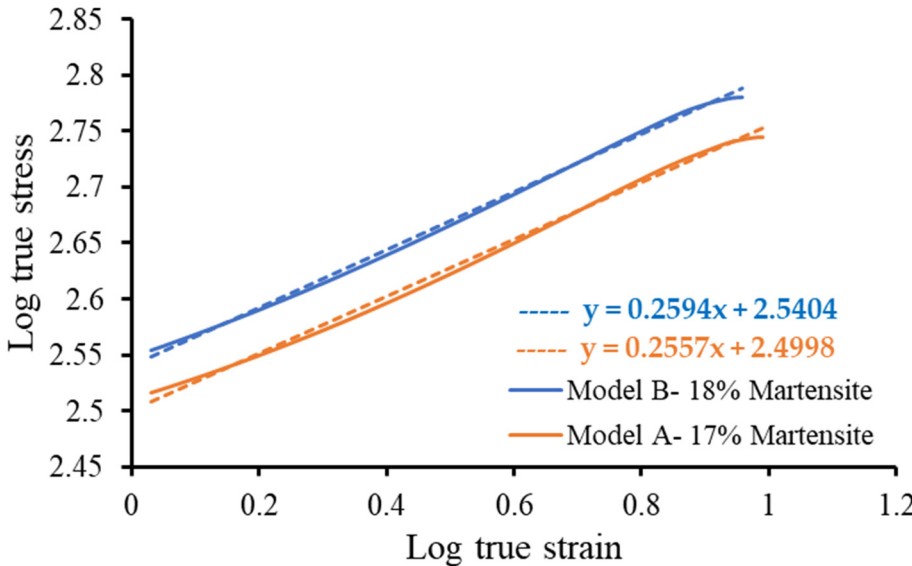

**Figure 6.** Comparison of strain hardening exponent (n) with corresponding martensite volume fraction, i.e., 18% and 17%.

**Table 4.** N-values versus martensite fractions of DP steels.

| n-Values | Martensite Fractions |
|---|---|
| 0.2594 | 18% |
| 0.2557 | 17% |

### 3.2. Effect of Martensite Grain Size on Stress–Strain Curve and FLDs

By simulations of models (A, C, and D) as per Figure 2, the influence of the difference in martensite grain sizes was plotted as stress–strain curves of the four loading conditions in Figure 7. The stress–strain curves do not significantly affect the variation of martensite grain size in uniaxial loading. Contrarily, it was observed that the maximum stress of medium and small martensite grain size was almost the same, but the RVEs with bigger martensite sizes failed at lower strain values.

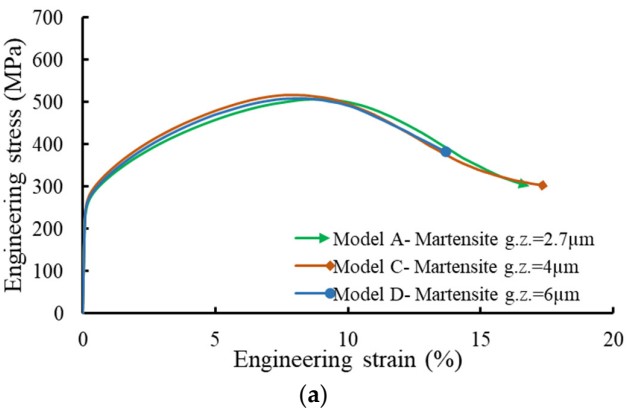

(a)

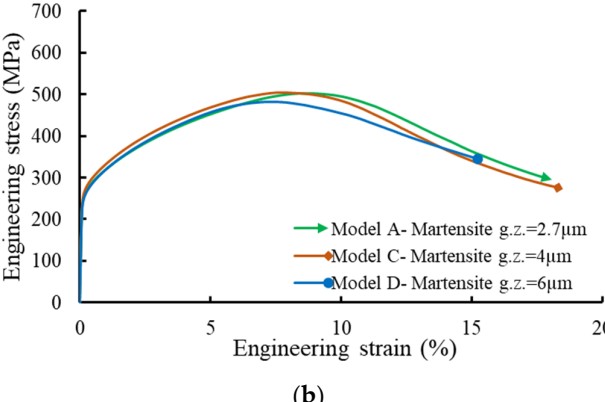

(b)

**Figure 7.** *Cont.*

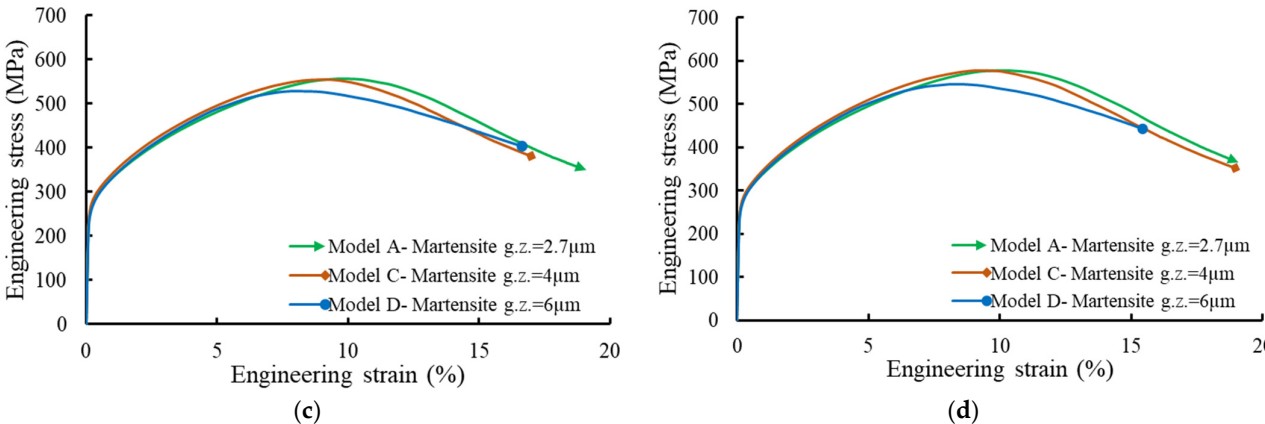

**Figure 7.** Comparison of stress–strain curves of martensite grain sizes in (**a**) uniaxial tension, (**b**) plane strain tension, (**c**) partial biaxial tension, and (**d**) equi-biaxial loading.

By increasing martensite grain sizes, the limiting major and minor strains decreased in the case of the M-K approach with some exceptions in uniaxial loading cases, as shown in FLDs in Figure 8. On the other hand, in the case of the K-B approach, the variation in martensite grain sizes did not have a remarkable effect on formability.

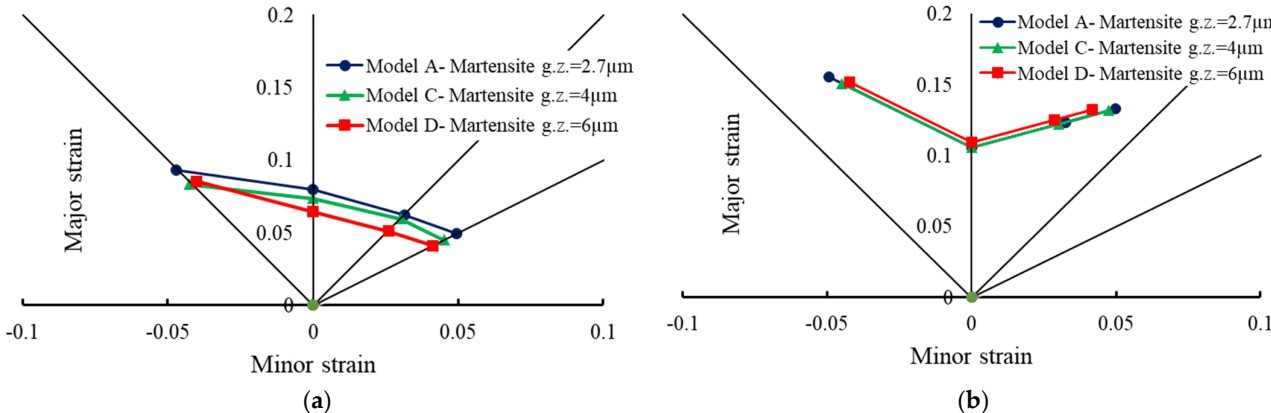

**Figure 8.** The FLDs in martensite grain sizes in the case of (**a**) M-K approach and (**b**) K-B approach.

### 3.3. Necking Band

In the case of the M-K approach, as shown in Figure 9, the FLD band was observed for models A and B, where the green line is the necking start at 0% of material degradation, the blue line is localized necking point, and the red line represents the fracture of the RVE at 20% of material degradation. The area under the green line is a safe zone, and above the blue line is the damage zone, where the necking starts.

While in the case of K-B, by using models A and D, the values of major strains at the plane strain loadings were almost the same when the damage values were changed, while the values of the major and minor strains in other loading conditions changed. In addition, the marginal zone of the FLD was not observed for either model, as shown in Figure 10.

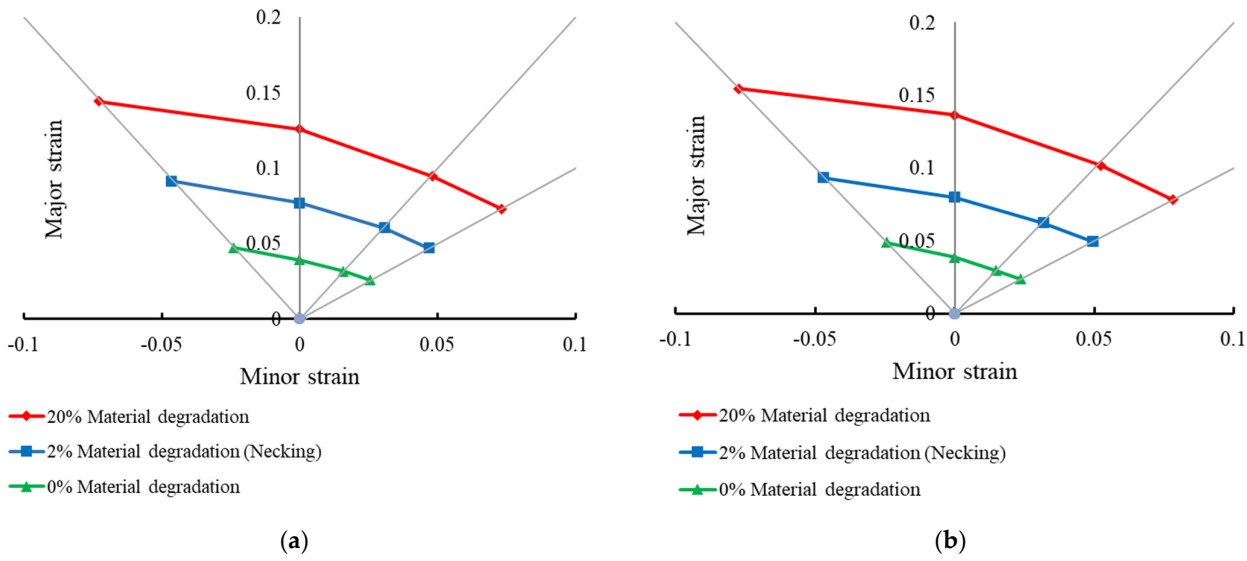

**Figure 9.** FLDs are showing marginal zones of (**a**) model A and (**b**) of model B.

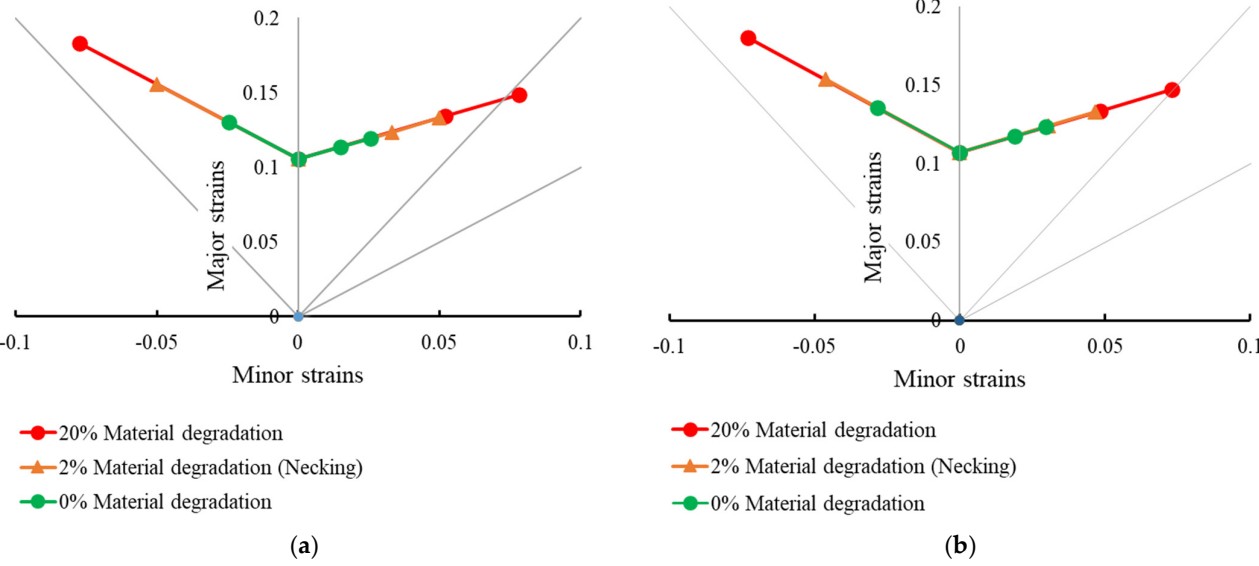

**Figure 10.** The FLDs marginal zones for (**a**) martensite grain sizes = 2.7 μm and for (**b**) martensite grain sizes = 6 μm.

### 3.4. Local Damage Evolution

Damage initiation in models A and C were analyzed on the top surfaces of 3D RVEs, as shown in Figures 11 and 12, respectively. The frames were extracted during post-processing of the simulation results to compare local behaviors of the RVEs at the necking point, which showed 20% and 30% of global material degradation, respectively. The local damage field is presented with the help of values ranging from 1.0 to 0.0, i.e., undamaged to fully damaged state, respectively.

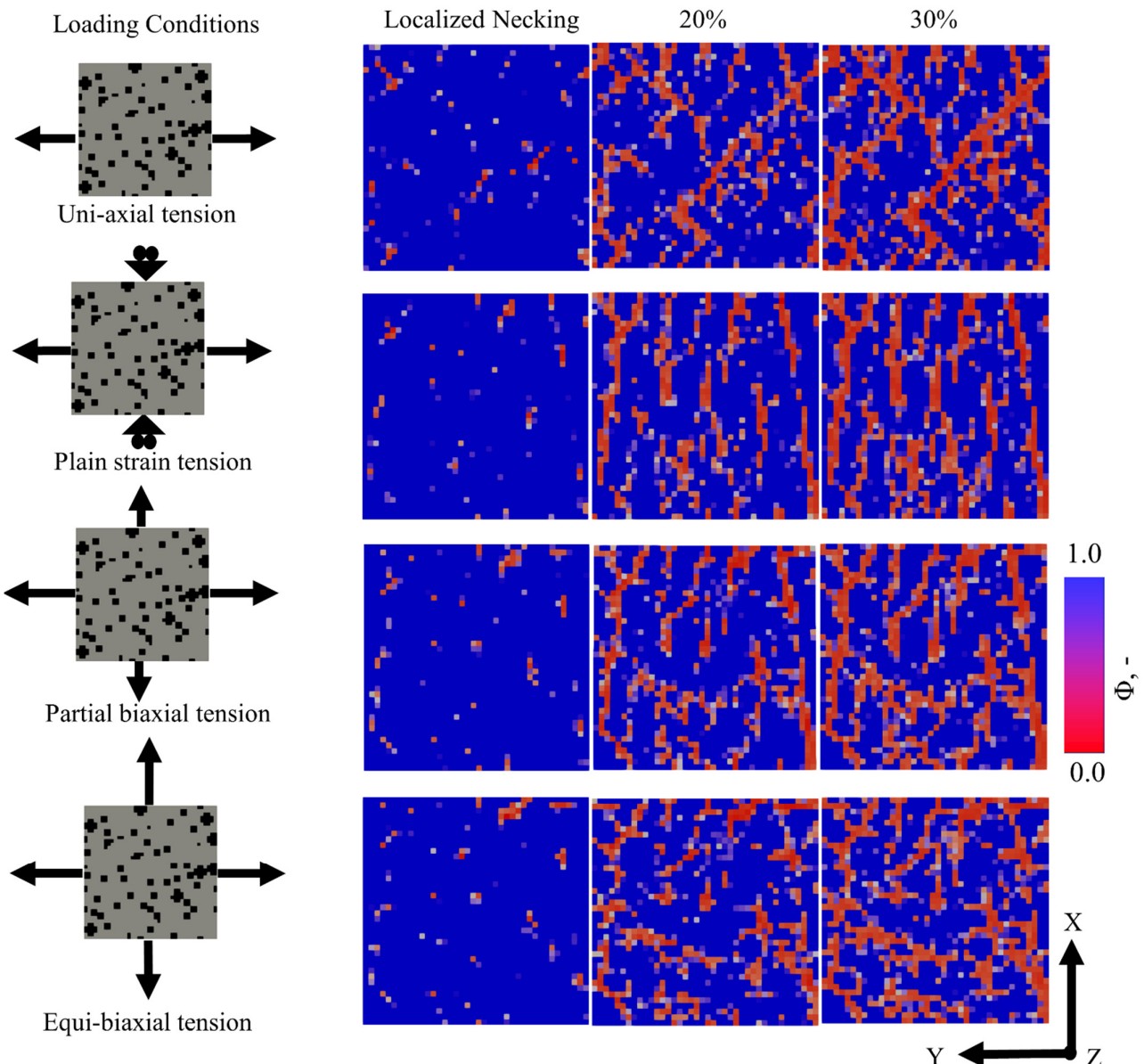

**Figure 11.** Damage behavior of dual-phase steels in model A with martensite grain sizes = 2.7 μm.

In model A (refer to Figure 11), the voids coalesced at 45 degrees of the load direction for uniaxial tension. The voids propagated sharply in this direction by the application of further load. In relation to plane strain loading, the damage initiation occurred at 45 and 90 degrees of load direction. With additional loading, the voids propagated as sharp straight lines at 90 degrees because of fixed loading at x-directions.

At partial biaxial loading, the damage was initiated by large voids. These voids propagated in 45 and 90 degrees to load directions. In contrast, in the equiaxial loading case, the damage propagated in different and random directions, such as 0, 45, and 90 degrees to load directions (refer to Figure 11). In the case of model C (refer to Figure 12), the local damage evolution did not differ much from model A, especially in the uniaxial case. However, it was observed in the plane strain case that the larger martensite grains constraint the damage behavior, and the damage tended to form around the martensite grains. The damage propagation also behaved in the same manner in the case of partial-biaxial and equi-biaxial loadings. The damage was initiated at the inter-granular level but continued growing trans-granularly with increased matrix degradation. In addition, stress relaxation near the damage areas affected the damage propagation and strain localization.

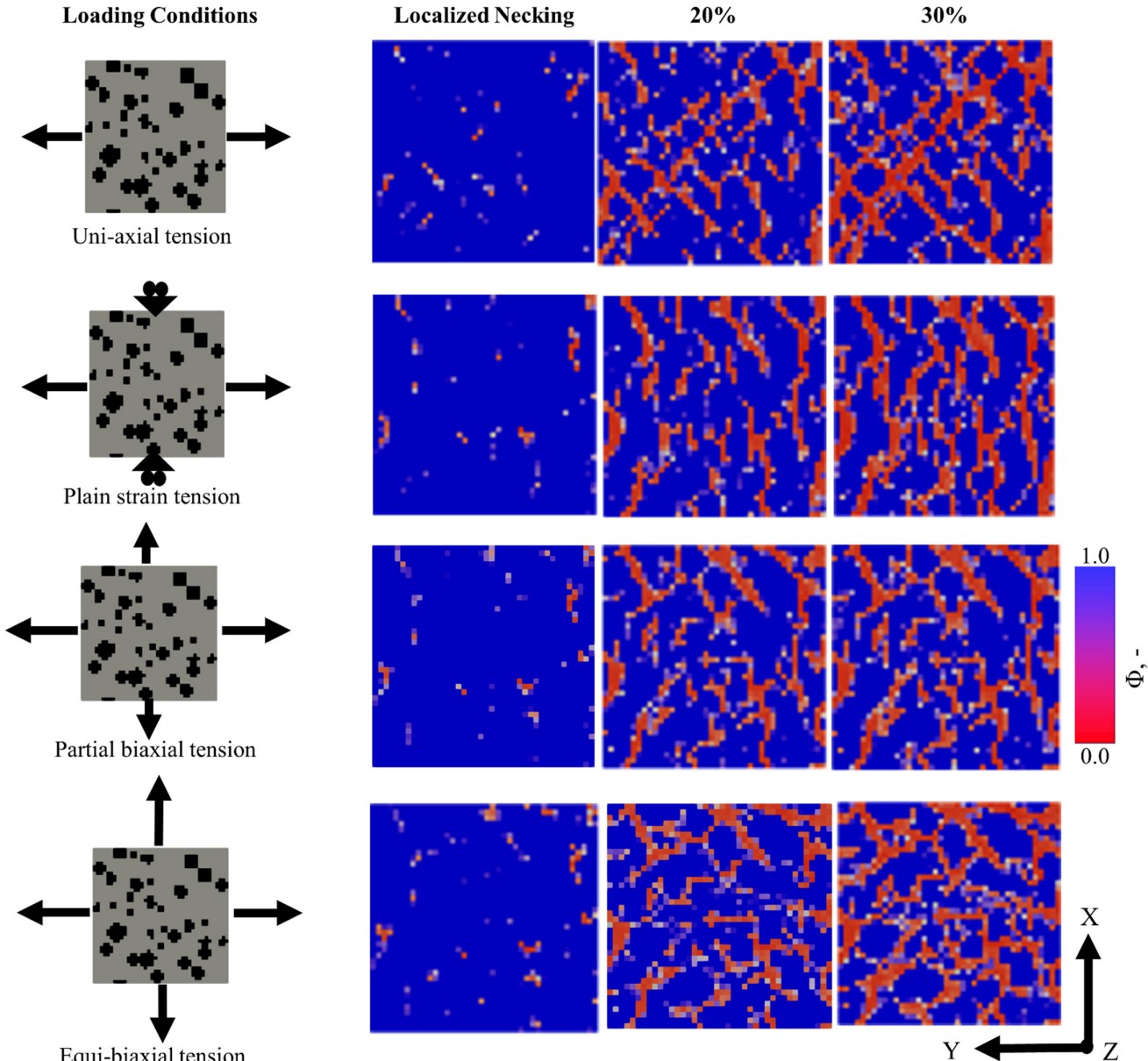

**Figure 12.** Damage behavior of dual-phase steels in model C with martensite grain sizes = 4 μm.

## 4. Discussion

Most analytical models used for plotting FLDs detect the forming limits at the necking point of materials, but calculating major and minor limit strains by numerical simulations is still required. In this study, these limiting strain values were detected from logarithmic strain tensor components, where the maximum value inside the tensor matrix represents the major strain, and minimum value is minor strain. Logarithmic strain is widely used in numerical simulations and is the best indicator for true strain, as its ability to count the strain values at every time interval and gives us an exact number when deformation occurs by a series of increments [31]. The strain tensor of this matrix represents an imaginary square unit, where minor strain is the minimum logarithmic change value in the unit square, and major strain is the maximum logarithmic change in this unit.

The increasing value of ultimate tensile strength by increasing phase fraction of the martensite is because of the enhanced reinforcement by uniformly distributed hard martensite particles [32]. On the other hand, the reduced capability of the material to undergo global strains with more martensite fraction is due to the brittle nature of martensite and

the comparatively less force shifted to the ferrite grains. Furthermore, the martensite's decreased grain size provokes the surrounding ferrite's plastic deformation; therefore, the DP steel sample with a small martensite grain size showed comparatively better formability [33].

The behavior of FLDs agrees with the results of Duancheng Ma [30], where it applies on the right side of FLD with and without periodic boundary conditions, as shown in Figure 5 for the M-K approach. On the other hand, in the K-B approach, the simulations, as shown in Figure 5b, agree with Basak [8], where results are presented for DP steels 600, 800, and 1000, and G Béres [12], who worked on aluminum alloy Al 2008-T4. In Figure 5a, the FLDs with a relative difference in trends influenced by martensite fraction compared to those in Figure 5b, where no significant difference was observed, establish the argument that the M-K approach corresponds better to the crystal plasticity-based modeling approach. Furthermore, this effect was observed to be more pronounced in the case of using the M-K approach to evaluate the influence of variation in the martensite grain size, as shown in Figure 8.

Keeler–Brazier's equations did not illustrate the difference of formability by changing martensite grain sizes. Furthermore, they did not show a significant necking band, as the damage values at plane strain loading conditions were not altered, as shown in Figures 8 and 10. On the other hand, the damage values' dependency on engineering stress–strain curves can clarify the 0.2% proof stress points, onset of necking, and fracture for the M-K approach.

In addition, n-values were calculated for both models using Equation (6).

$$\sigma_t = k\,\varepsilon_t^n \tag{6}$$

where $\sigma_t$ is the true stress, k is a constant, $\varepsilon_t$ is the true strain, and n is the strain hardening coefficient. These n-values can keep their higher strength values in the pre-necking zones, reducing the risk of local strain accumulations and uniformly distributing the strain over the whole domain, improving materials' formability. True stress–strain curves of DP steels and the plastic work performed by biaxial and uniaxial loading using crystal plasticity models show a good agreement with Equations (7) and (8) used during experimental work.

$$\sigma_t = \sigma_e\,(\,1+\varepsilon_e\,) \tag{7}$$

$$\varepsilon_t = \ln\,(\,1+\varepsilon_e\,) \tag{8}$$

where $\sigma_t$ is the true stress, $\varepsilon_t$ is the true strain, $\sigma_e$ is the engineering stress, and $\varepsilon_e$ is the true strain. These equations are based on the ISO 16842 standard for studying biaxial tensile testing on sheet metals [34].

The local damage behavior of the DP steel samples upon varying loading conditions showed different outcomes, which can help understand the corresponding effect of microstructure. This is also affected by the morphology of the martensite particles, their orientation, aggregation, and presence in small islands. The initiation and propagation of local damage at the martensite–ferrite grain interface are because of the decohesion of the comparatively weak point in the aggregate [35,36]. The direction of the damage propagation in the case of tensile loading in ductile ferrite matrix, i.e., at 45 degrees, is caused by the shear bands [29]. A further extensive study is needed to understand this effect by considering the contributing factors.

## 5. Conclusions

The dependence of microstructural attributes on the formability limit of DP steels was analyzed. Specifically, the averaged global behavior and the effect of different loading conditions on damage initiation were investigated. Several 3D RVEs with varying microstructural attributes were simulated using a crystal plasticity-based numerical simulation model called DAMASK. The global and local stress, strain, and damage evolution

of various RVEs revealed the internal phenomena during deformation. The global results were processed to obtain FLDs according to K-B and M-K criteria. Appealing outcomes were observed, which can be summarised by the following points.

1.  DAMASK can model FLDs of multi-phase materials with different loading conditions. Therefore, it can be used to study the effect of microstructural attributes on the formability of crystalline materials.
2.  There are some limitations in the K-B model; firstly, the equations did not introduce a significant difference in FLDs by using different martensite grain sizes. Secondly, they did not possess the traditional marginal zones of FLDs. Contrarily, the M-K approach was proved to have good efficacy and agreement with the previous study of Duancheng Ma [7]. Moreover, it can clarify differences in mechanical behavior influenced by varying grain sizes of martensite in FLDs. Consequently, it shows comparatively safe necking and damage zones for different loading conditions.
3.  In plotting FLDs, it was found that the lower the martensite fractions in DP steels, the better the formability. The precipitates of martensite act as obstacles, which restricts the slip deformation. Regarding martensite grain size in DP steels, the formability generally improves by decreasing martensite grain size. The higher strain hardening coefficient values (n-values), the better the formability because of comparatively low martensite phase fractions, which is a hard phase.
4.  Plastic work and strength values of biaxial loading cases are higher than those in uniaxial and plane strain loading. At the same time, the plastic work values increase by decreasing martensite fractions in DP steels.
5.  The difference in martensite grain sizes and loading conditions strongly affects the damage initiation and propagation behaviors of the RVEs, which could serve as a good guide on how to avoid damage propagation in the future.

**Author Contributions:** Conceptualization, T.H. and F.Q.; methodology, T.H.; software, T.H., F.Q. and M.U.; validation, T.H., F.Q. and S.G.; formal analysis, M.U.; investigation, T.H. and F.Q.; resources, F.Q. and S.G.; data curation, T.H.; writing—original draft preparation, T.H., M.U. and F.Q.; writing—review and editing, F.Q., M.U. and S.G.; visualization, T.H.; supervision, F.Q. and S.G.; project administration, S.G. and U.P.; funding acquisition, T.H. and U.P. All authors have read and agreed to the published version of the manuscript.

**Funding:** This research received no external funding.

**Institutional Review Board Statement:** Not applicable.

**Informed Consent Statement:** Not applicable.

**Data Availability Statement:** The simulation data are not publicly available but can be shared upon request.

**Acknowledgments:** The authors acknowledge the DAAD Faculty Development for Candidates (Balochistan), 2016 (57245990)-HRDI-UESTP's/UET's funding scheme in cooperation with the Higher Education Commission of Pakistan (HEC) for sponsoring the stay of Faisal Qayyum at IMF TU Freiberg. This work was conducted with the DFG-funded collaborative research group TRIP Matrix Composites (SFB 799). The authors gratefully acknowledge the German Research Foundation (DFG) for the financial support of SFB 799. Furthermore, Freunde und Förderer der TU Bergakademie Freiberg e.V. is acknowledged for providing financial assistance to Muhammad Umar. The authors also acknowledge the support of Martin Diehl and Franz Roters (MPIE, Düsseldorf) for their help regarding the functionality of DAMASK. Finally, the competent authorities at Khwaja Fareed University of Engineering and Information Technology, (KFUEIT) Rahim Yar Khan, Pakistan, and TU BAF Germany are greatly acknowledged for providing research exchange opportunities to Muhammad Umar at the Institute of Metal Forming TU BAF, Germany, under a memorandum of understanding (MoU).

**Conflicts of Interest:** The authors declare no conflict of interest.

**Nomenclature**

**Acronym**

| Symbol | Description |
|---|---|
| CPFEM | Crystal plasticity finite element method |
| DAMASK | Düsseldorf Advanced Materials Simulation Kit |
| DP | Dual-phase steels |
| FEM | Finite Element Method |
| FFT | Fast Fourier Transform |
| FLD | Forming Limit Diagram |
| K-B equations | Keeler–Brazier equations |
| M-K approach | Marciniak and Kuczynski approach |
| RVE | Representative volume element |
| SP | Spectral method |
| g.z. | Grain Size (μm) |

**Appendix A.**

*Appendix A.1. Running Simulation via CP Spectral Solver*

The numerical simulations were processed by incorporating ductile damage and recording more increments after the damage initiation than those without damage in each RVE. During numerical processing by the spectral method, each grid point inside the mesh of the RVE acts as a computation point and represents individually defined deformation mechanisms, phase fractions, grain orientation, and homogenization schemes. Each increment after damage initiation records the material degradation behavior in detail. Once the damage is initiated, the numerical processing slows down and becomes intensively computed, and crashes after specific material degradation occurs for the RVE. The simulations and the load increments in each loading condition in this work were processed till converging.

*Appendix A.2. Postprocessing Stage*

The completed numerical simulations were post-processed using customized subroutines on DAMASK [37]. FLDs were plotted by adopting major and minor strains. The following two approaches are generally accepted, and commonly employed techniques based on some equations and engineering stress and strain curves to adopt the major and minor forming limit strains as forming limit criteria.

Appendix A.2.1. M-K Approach

For the M-K approach, after plotting engineering stress and strain curves, the maximum stress (localized necking point) was detected, and the strain tensors were recorded along with the increments. Major and minor values were detected from the strain tensor matrix at an increment, corresponding to this maximum stress in the stress–strain curve, as shown in the following process chart (see Figure A1).

Appendix A.2.2. K-B Approach

The Keeler-Brazier model depends on the major true strain value FLD0, true on the thickness of the virtual samples, and the strain hardening coefficient when the minor true strain equals zero [13]. It presents Equations (A1)–(A3) to predict the FLDs following the activities, as shown in Figure A2, where major true strain $\varepsilon_1$ is dependent on values of thickness t and strain hardening coefficient n.

For $\varepsilon_2 < 0$ (uniaxial case)

$$\varepsilon_1 = \text{FLD}_{0,\ \text{true}} - \varepsilon_2 \tag{A1}$$

For $\varepsilon_2 > 0$ (biaxial case)

$$\varepsilon_1 = \ln[0.6 \times (\exp(\varepsilon_2) - 1) + \exp(\text{FLD}_{0,\ \text{true}})] \tag{A2}$$

where,

$$\text{FLD}_{0,\,\text{true}} = \ln[1 + (0.233 + 0.413xt) \cdot n/t] \qquad (A3)$$

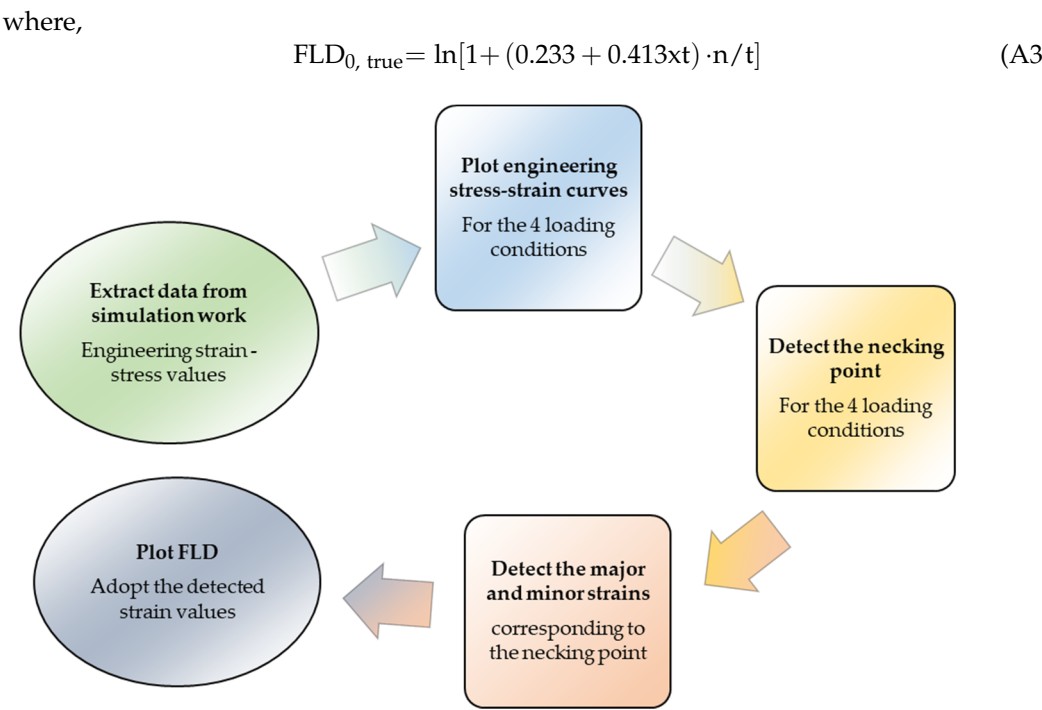

**Figure A1.** Process chart of plotting FLD via M-K approach.

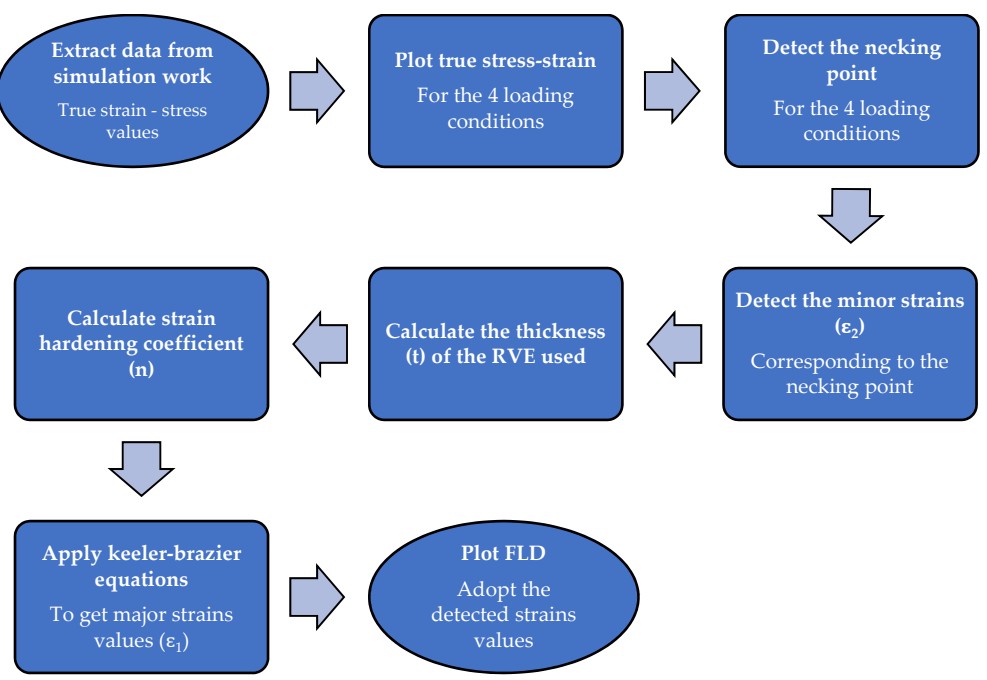

**Figure A2.** Process chart of plotting FLD via Keeler-Brazier (K-B) approach.

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
