# Peer review of "Micromechanical Effect of Martensite Attributes on Forming Limits of Dual-Phase Steels Investigated by Crystal Plasticity-Based Numerical Simulations"

_crystals, doi:10.3390/cryst12020155_

Round 1

Reviewer 1 Report

1- The abstract should be revised and core finding of the research should be added.

2- How the authors verify the simulation results. It should be clearly explained.

3- In Figure 7, it seems that the martensite percentage and its effect on the FLD diagrams is not interpreted clearly.

4- The discussion on the results should be more explained.

Author Response

Detailed document attached in .dox format.

Reviewer 2 Report

This manuscript nicely described the effect of microstructural attributes on forming of dual-phase steels using a numerical simulation. The research is interesting and can be acceptable for the publication after minor revisions following the comments below. 

C1. The purpose and significance of the research are not clear from this title. The title is needed to revise.

C2. Novelty of the works should be clearly focused/mentioned in the abstract.

C3. Also, in this study, the novelty is not clear from the Introduction.

C4. Use of many Figures/data, the MS looks very lengthy or not suitable for reading! Some of the explanations should be concise.

C5. Less important data/Figures should be shifted in the Supporting Information.

Reviewer 3 Report

I hope that the authors will develop equation 9 further in future articles and present the physical meaning of the coefficients k and n.

Round 2

Reviewer 1 Report

The manuscript can be accepted in the present form